# Relationships between Depression, Fear of Childbirth, and Obsessive-Compulsive Symptoms among Pregnant Women under the COVID-19 Pandemic in Japan

**DOI:** 10.3390/healthcare11030361

**Published:** 2023-01-27

**Authors:** Yuriko Usui, Mizuki Takegata, Satoru Takeda, Toshinori Kitamura

**Affiliations:** 1Department of Midwifery and Women’s Health, Division of Health Sciences and Nursing, Graduate School of Medicine, The University of Tokyo, Tokyo 113-0033, Japan; 2Kitamura Institute of Mental Health Tokyo, Tokyo 151-0063, Japan; 3Department of Obstetrics & Gynecology, Faculty of Medicine, Juntendo University, Tokyo 113-8421, Japan; 4Aiiku Research Institute for Maternal, Child Health and Welfare, Imperial Gift Foundation Boshi-Aiiku-Kai, Tokyo 106-0047, Japan; 5Kitamura KOKORO Clinic Mental Health, Tokyo 151-0063, Japan; 6T. and F. Kitamura Foundation for Studies and Skill Advancement in Mental Health, Tokyo 151-0063, Japan; 7Department of Psychiatry, Graduate School of Medicine, Nagoya University, Nagoya 464-0814, Japan

**Keywords:** antenatal depression, COVID-19, personality, adult attachment, fear of childbirth, obsessive-compulsive disorder

## Abstract

Little is known about the causality of antenatal depression (AND). We focused on the causal relationships between AND, fear of childbirth (FOC), and obsessive-compulsive symptoms (OCS) during the COVID-19 pandemic in Japan. We also examined whether the perceived threat of COVID-19 is associated with AND. Participants were recruited for an Internet survey conducted in December 2020. A total of 245 pregnant women completed the online survey at 12 to 15 weeks’ gestational age (Time 1) and approximately 10 weeks later (Time 2). AND was estimated using the first two diagnostic items of Major Depressive Episode. The estimated prevalence of AND was 4.5% and 2.9% at Time 1 and Time 2, respectively. At both time points, no association was found between AND and the perception of COVID-19 threat. Structural equation modeling showed that AND predicted OCS (β = 0.16, *p* < 0.001), which, in turn, predicted FOC (β = 0.09, *p* = 0.042); FOC, in turn, predicted AND (β = 0.23, *p* < 0.001). AND, OCS, and FOC were predicted by borderline personality traits. Insecure adult attachment influenced AND and FOC via the perceived negative impact of the current pregnancy. Perinatal care providers should assess the personality and perception of pregnancy to prevent depression and pay attention to symptoms such as FOC and OCS in addition to those of depression.

## 1. Introduction

Depression often occurs among pregnant women. The pooled prevalence of AND identified by a meta-analysis was 20.7%, and that of major depression was 15% [1]. Several correlates of AND in previous studies were identified: primiparity, single motherhood, history of mental illness, unplanned and/or unwanted pregnancy, pregnancy complications, past history of abortion, early life experience such as loss of father, vulnerable personality such as neuroticism and anxiety traits, being exposed to domestic violence, lack of social support, and poor intimacy with the husband [2,3,4,5]. It is of note that these variables are not mutually independent.

Out of the correlates, poor relationship with the partner or being exposed to domestic violence would be strongly contributing to the onset of perinatal depression [2,3]. Considering pregnancy is a life stage in which interpersonal relationships, particularly with a partner, play a critical role preparing for childbirth and child rearing [6]. The person whom pregnant women seek support the most is her partner [7], and the partner’s support contributes to the mental health of pregnant women [8]. Although poor interrelationship is caused by the malfunctional interaction between individuals with different personality [9], it is speculated that a woman with an insecure attachment style would underlie the poor relationship among the couple, leading to the occurrence of AND.

Furthermore, insecure attachment to a partner may be associated with maladaptive personality traits. Special attention may be necessary for individuals with borderline personality traits. Borderline personality disorder, which occurs in approximately 1.4% of the general population [10], is one of the most prevalent personality disorders [11]. People with borderline personality disorder are often accompanied by a variety of mood and anxiety disorders. Hence, there may be a strong association between insecure adult attachment to a partner, borderline personality, and AND.

Recent research indicates that fear of childbirth (FOC) is frequently observed among pregnant women. Reportedly, 6–14% of pregnant women expressed severe FOC [12]. Women with FOC are afraid of pain, unpredicted complications such as bleeding, and uncontrollable situations [13,14] (Nilsson & Lundgren, 2009; Takegata et al., 2018). Studies have reported that FOC and depression coexist [15,16,17]; hence, FOC should not be dismissed when considering the correlates of AND. 

Furthermore, a recent systematic review and meta-analysis reported that the onset of obsessive-compulsive disorders (OCDs) during pregnancy is not uncommon, and that 13–39% of women with OCD develop the disorder during their pregnancy [18]. Depression frequently accompanies obsessive-compulsive symptoms (OCS) [19,20]. The coronavirus 2019 (COVID-19) pandemic may worsen these fears and obsessions in pregnant women who are already affected by FOC or OCS, and their aggravation may affect AND. Hence, our primary interest was to investigate the causal relationships between AND, FOC, and OCS among pregnant women. We were also interested in the effects of adult attachment style and borderline personality traits on the above relationships.

The COVID-19 outbreak began worldwide in February 2020. In Japan, most obstetric facilities canceled or restricted perinatal classes with the COVID-19 pandemic; thus, pregnant women were less likely to obtain support from healthcare providers. Since the outbreak of COVID-19, reports have been inconsistent in terms of the prevalence of perinatal depression during the pandemic: some have shown an increased prevalence of perinatal depression [21,22,23,24,25,26,27,28,29,30,31,32,33,34,35,36,37,38,39,40], whereas others have refuted this or have reported a decrease in depression [41,42,43,44,45]. Disastrous events, such as the COVID-19 pandemic, are often followed by psychological maladjustment of individuals exposed to them (e.g., [46]). A risk factor related to disaster-induced psychological maladjustment is perceiving the event as having adversely affected one’s health. Residents in an area near a nuclear power plant accident (Three Mile Island; TMI) showed no increase in episodes of depression or anxiety as compared with those living in an area near a nuclear plant without an accident. However, those who manifested an episode of depression or anxiety, as compared with those who did not, were more likely to view the situation at TMI as currently dangerous, believe that living near a nuclear facility was unsafe, and had young children in the house [47]. Hence, another research question was whether the perceived threat of COVID-19 is associated with AND among the Japanese population. 

## 2. Methods

### 2.1. Study Procedure and Participants

This Internet survey was conducted with pregnant women at 12–15 weeks’ gestational age (Time 1: T1). No exclusion criteria were used except for a lack of command on Japanese reading. The T1 survey was conducted for two weeks, from 7 to 21 December 2020, via the Internet applications called Luna Luna and Luna Luna Baby (MTI Ltd., Tokyo, Japan). Anonymity was assured, and participation was voluntary. The main questionnaire was preceded by an information page, with the aims of the study, affiliations, information about informed consent, and the address of the consultation desk for the research. Participants were asked to enter their email address after completing the survey to merge the data of T1 and T2 only by means of mail address. The participants were incentivized with electronic money points usable for Amazon shopping via the MTI Ltd (MTI Ltd., Tokyo, Japan). 

The same participants were invited to participate in the Time 2 (T2) study approximately 10 weeks later (22–35 gestational weeks) in March 2021. As compared with the women who declined to participate in the T2 survey, those who participated in the T2 survey were slightly but significantly older (32.2 vs. 31.4 years, *p* < 0.05), and their gestational age was less advanced (13.2 vs. 13.4, *p* < 0.05). The two groups of women did not significantly differ otherwise. The period was at the beginning of the fourth wave of the COVID-19 pandemic. The same questionnaires were used, except for basic demographic and obstetric variables. Because this Internet survey set the items as forced questions, there was no missing value.

### 2.2. Measurements

Depression: The first two symptoms of Major Depressive Episode (MDE)―depressed mood and lack of interest—were observed to detect depression. Each item was rated on a 4-point scale: 0 = none, 1 = a few days a week, 2 = more than half a week, and 3 = almost every day. This was based on research showing that the set of two questions would predict MDE reasonably well [48,49,50,51,52,53,54,55]. As a rough indicator of MDE, we defined MDE at both time periods if either or both of the two items (depressed mood and lack of interest) were rated as present almost every day for the previous 2 weeks. In this study, Cronbach’s α coefficients were 0.83 at T1 and 0.79 at T2.

Borderline personality traits: The participant’s borderline personality trait was rated on the short version (IPO-SV) [56] of the Personality Organisation Inventory (IPO) [57]. The IPO-SV has nine items on a 7-point scale with three subscales: primitive defense (PD), identity diffusion (ID), and reality testing (RT) disturbance. In this study, Cronbach’s α coefficients of the PD, ID, and RT subscales were 0.71, 0.78, and 0.87, respectively.

Adult attachment: The participants’ attachment toward their partners was rated using the Japanese version [58] of the Relationship Questionnaire (RQ) [59]. The RQ has four items answered on a 7-point scale. The four items describe different styles of adult attachment: Secure, Fearful, Preoccupied, and Dismissing. The total score was calculated by adding the scores of Fearful, Preoccupied, and Dismissing, which was then subtracted by the score of Secure. A higher score indicates a more insecure attachment style. In this study, Cronbach’s α coefficient was 0.65.

Perceived impact of pregnancy: The participants’ perception of the impact of the current pregnancy was rated by a single ad hoc item; ‘Mark a point for the influence of the current pregnancy upon you from −100 to 100.’ and ‘Mark positive points if it is good, joyful, and happy, whereas mark negative points if it is awful, perplexing, and unhappy.’

Demographic and obstetric variables: We asked the participant’s age, number of past pregnancies, number of past deliveries, infertility treatment, and marital status (single/married).

Antenatal fear of childbirth: We used the Japanese version [60] of the Wijma Delivery Expectancy/Experience Questionnaire [61]. It consists of 33 items rated on a 5-point scale. A higher score indicates more severe FOC. Item 31 was erroneously deleted in the present study. In this study, Cronbach’s α coefficients were 0.91 at T1 and 0.91 at T2.

Obsessive-compulsive symptoms: Obsessive and compulsive symptoms were rated using the Japanese version [62] of the Obsessive-Compulsive Inventory—Revised (OCI-R) [63]. The OCI-R comprises 18 items with a 5-point scale. In this study, we changed the grading from 5-point to 7-point. In this study, Cronbach’s α coefficients were 0.89 at T1 and 0.90 at T2.

Perceived threat of COVID-19: We used ad hoc questions. This was rated using three items on a 7-point Likert scale from not at all true = 0 to very much true = 6: “potentially dangerous”, “not serious unless aged or with choric disease (reverse)”, and “OK because I am immune (reverse)”. In this study, Cronbach’s α coefficient was 0.67.

### 2.3. Data Analysis

After examining the means, standard deviations (SDs), skewness, and kurtosis of all the variables studied in the present investigation, we correlated the T1 and T2 depression scores with the scores of the other explanatory variables. We set the alpha value to *p* < 0.001 due to multiple comparisons. Then, using the variables that were significantly correlated with T1 and T2 depression scores, we set a non-recursive structural regression model in the framework of structural equation modeling (SEM) (Figure 1). Here, we posited four latent variables: borderline personality traits, adult attachment, and two depressions at T1 and T2. We hypothesized that OCS, FOC, and depression at T1 were all predicted by borderline personality traits and adult attachment that were mediated by the perceived impact of the present pregnancy. Borderline personality traits and adult attachment were correlated. OCS, FOC, and depression at T1 were correlated with each other, and all predicted their own scores at T2. At T2, OCS, FOC, and depression predicted each other, making a non-recursive loop (Figure 1). The goodness of fit of this model was rated by χ^2^, comparative fit index (CFI), and root mean square error appropriation (RMSEA). A good fit was defined as χ^2^/*df* < 2, CFI > 0.97, and RMSEA < 0.05. An acceptable fit was defined as χ^2^/*df* < 3, CFI > 0.95, and RMSEA < 0.08 [64,65]. After confirming that the model fit was good, we performed model trimming in which the least significant path was deleted one by one until (a) no non-significant path remained to be deleted, (b) the model’s solution was inappropriate, or (c) the model fit was no longer good. Statistical analyses were performed using IBM SPSS version 28.0 and Amos version 28.0 software for Windows (IBM Japan, Tokyo, Japan).

### 2.4. Ethical Considerations

This study was approved by the Institutional Review Board (IRB) of the Kitamura Institute of Mental Health, Tokyo (No. 2020101501). All participants provided electrical informed consent after understanding the study rationale and procedure. The authors assert that all procedures that contributed to this study complied with the ethical standards of the National and Institutional Committees on Human Experimentation and with the 1975 Declaration of Helsinki as revised in 2008.

## 3. Results

Of the 696 pregnant women who participated in T1, data from 245 (35.2%) who participated in T2 were used in the analysis.

Although AND, FOC, and OCS were correlated with each other at both T1 and T2, none of them were correlated with the perceived threat of COVID-19 (Table 1).

The prevalence of MDE was 4.5% and 2.9% at T1 and T2, respectively. Of the 11 women identified as having MDE at T1, 9 (82%) still had MDE at T2. Of the 234 women who did not have MDE at T1, 5 (2.1%) had MDE at T2. T1 and T2 depression scores were both significantly correlated with borderline personality traits, insecure adult attachment, perceived impact of the current pregnancy, and T1 and T2 W-DEQ and OCI-R (Table 2).

The initial model showed a good fit: χ^2^/*df* = 1.351, CFI = 0.984, and RMSEA = 0.038. This persisted after model trimming: χ^2^/*df* = 1.337, CFI = 0.983, and RMSEA = 0.037 (Figure 2). The trimmed SEM model showed that T1 OCS, FOC, and depression were predicted by borderline personality traits. T1 FOC and depression were predicted by (insecure) adult attachment and mediated only by the (negative) perceived impact of the present pregnancy. At T2, depression was predicted using FOC. Depression then predicted OCS, which, in turn, predicted FOC. Hence, a circuit of prediction appeared. The direction of prediction was not found the other way round. 

## 4. Discussion

The use of a non-recursive model in which OCS, FOC, and depression were set to statistically predict each other was unique to this study. It is of note that our study examined the correlations between the target variables at one time point as well as between the two time points; we used "influence" and "predict" only as statistical terms but did not necessarily indicate real causality. We cannot be more cautious about unseen confounders and mediators. We found relationships from FOC to depression, depression to OCS, and OCS to FOC. The directions the other way round were not significant. The W-DEQ comprises four components, one of which is isolation. Isolation is a risk factor for depression, especially during the COVID-19 pandemic, where it is difficult to have contact with other people [66]. Care for women with FOC, including pregnant women who have feelings of isolation, may prevent depression. Additionally, the SEM showed that depression predicted OCS. Since OCS at T2 was strongly influenced by T1 OCS, we interpret that persistent OCS stems from the first trimester, which is further aggravated by depression. Therefore, this result suggests that healthcare providers should carefully assess depression together with FOC and OCS, and consider how to approach expectant women with them taking into account the directions in which they affect each other.

Our results showed that borderline personality traits directly predicted AND. A recent meta-analysis estimated the prevalence of borderline personality traits during pregnancy in non-clinical samples to be 6.9–26.7% [67]. The existence of borderline personality is not rare among pregnant women, indicating the importance of identifying women with borderline personality before and early in pregnancy, and of preventing and caring for depression. Furthermore, borderline personality predicted not only depression but also FOC. A previous study reported that pregnant women with borderline personality traits expected childbirth to be traumatic and requested to give birth early [68]. In the perinatal period, it is important to care for women for not only depression but also FOC and OCS; for this, it is essential to assess their personality.

Although previous studies have reported that insecure attachment styles influence depression [69,70], this study found that insecure attachment styles are associated with more negative perceptions of pregnancy, which is more likely to lead to depression. In Japan, a birth debriefing generally provides women with an opportunity to talk about their birth experience with the midwife who assisted in the delivery to promote a positive perception of birth, but there are no routine opportunities to review the perception of pregnancy. Providing care that involves listening carefully to pregnant women about their perception of the current pregnancy is needed.

In this study, the prevalence of MDE was 4.5% and 2.9% at T1 and T2, respectively. Kitamura et al. (2006) conducted an epidemiological survey of 290 Japanese pregnant women using a structured diagnostic interview and reported a 5.9% incidence of Diagnostic and Statistical Manual of Mental Disorders—III-R (DSM-III-R) MDE in early pregnancy [71]. Considering that the current study relied only on the presence of two core symptoms of MDE at the time of investigation for diagnosing MDE, we assume that the prevalence of perinatal depression remained unchanged before and after COVID-19 or at least did not increase, as reported in other studies. Matsushima et al. (2022) reported that the point prevalence of AND was 17% among 1777 Japanese pregnant women [33]. The discrepancy between the study and our study may be due to the use of different scales used for identifying depression (the core two symptoms of DSM-5 and the Edinburgh Postnatal Depression Scale; EPDS). The EPDS comprises three factors: anhedonia, anxiety, and depression [72,73]. Therefore, it is likely that the pregnant women in the previous study scored high only on anxiety. A second reason may be the differences in assessment points; the research period of our study was in the middle of the second wave of the COVID-19 pandemic, whereas Matsushima et al.’s (2022) study was after the first wave. Therefore, perceived psychological stress/threats may have been desensitized during the pandemic period.

AND was not predicted by the perceived threat of COVID-19 in our study. Although in disastrous events such as a nuclear power station meltdown, the perceived threat is a predictor of psychological maladjustment, our expectant women seemed to be immune to the threat of infection. This may be due to the fact that the magnitude of the infection was much lower in Japan than in other countries.

There were limitations to this study. First, the scale to assess the threat of COVID-19 was ad hoc, and its validity and reliability have not yet been confirmed. Second, as this was an online study design, the response rate was low, and the characteristics of the sample might not be representative of all Japanese pregnant women. The incidence of AND was based on the presence or absence of the two core MDE symptoms. A more detailed assessment of all MDE symptoms is desirable. Regardless, the current study is the first to identify the circular causality between AND, FOC, and OCS, as well as the effects of borderline personality traits and insecure adult attachment on them among the Japanese population. This implies the importance of a full understanding of women’s personality traits, which precede the occurrence of mental disorders during pregnancy.

## 5. Conclusions

By means of a non-recursive SEM among Japanese pregnant women, we aimed to understand the causal relationships between borderline personality traits and adult attachment with FOC, OCS, and AND, as well as circular influences from AND through OCS to FOC and, in turn, predicting AND. However, no association was found between AND and the perception of COVID-19 threat.

## Figures and Tables

**Figure 1 healthcare-11-00361-f001:**
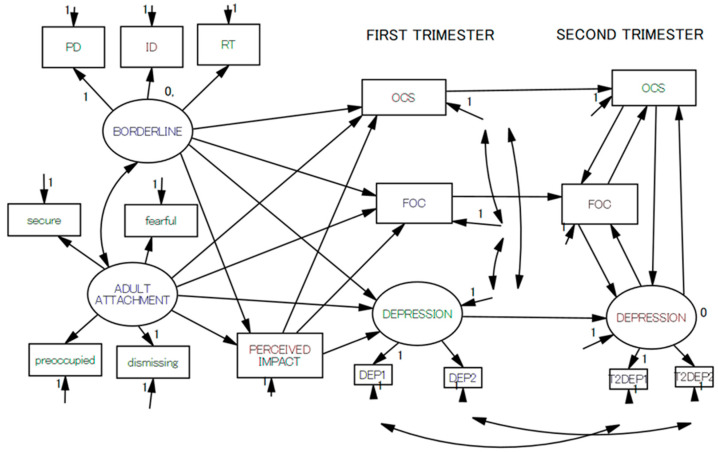
Conceptual model of causal relationships between antenatal depression.

**Figure 2 healthcare-11-00361-f002:**
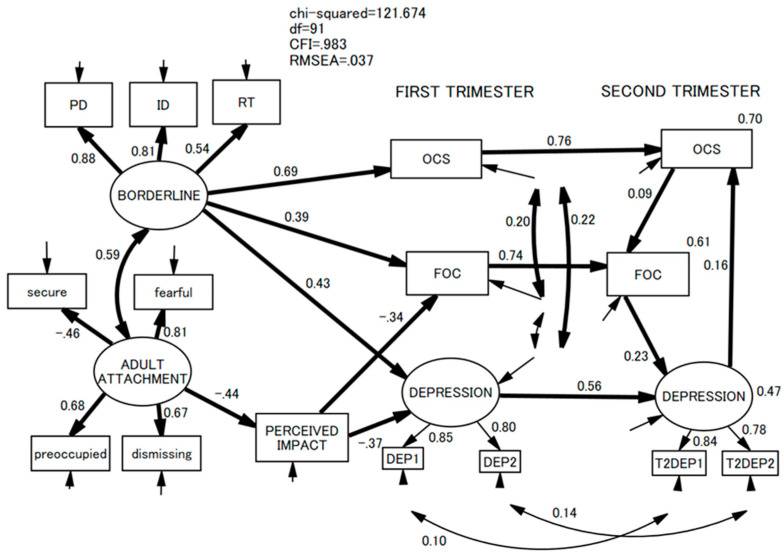
Final model after trimming.

**Table 1 healthcare-11-00361-t001:** Correlations of T1 and T2 depression (*N* = 245).

	Mean (*SD*) or *N* (%)	T1 Depression	T2 Depression
Demographic and obstetric variables			
Age	32.2 (4.2)	0.013	−0.009
Gestational age	13.2 (1.1)	−0.042	0.112
Gravidity	1.8 (1.1)	−0.175 *	−0.087
Primipara	131 (53.5)	---	---
Infertility treatment	85 (34.7)	---	---
Have a partner	243 (99.2)	---	---
Borderline personality traits			
T1 Primitive defense	4.44 (3.66)	0.403 *	0.352 *
T1 Identity diffusion	7.34 (4.73)	0.428 *	0.355 *
T1 Reality testing	1.58 (2.64)	0.231 *	0.332 *
T1 Total	13.4 (9.3)	0.440 *	0.412 *
Adult attachment			
T1 Total	−2.3 (3.9)	0.306 *	0.271 *
Perception of the impact of the current pregnancy			
Perceived impact of pregnancy	84.0 (27.7)	−0.452 *	−0.364 *
Fear of child birth			
T1 Total	62.0 (20.3)	0.428 *	0.395 *
T2 Total	66.0 (21.4)	0.358 *	0.425 *
Obsessive-compulsive symptoms			
T1 Total	27.4 (16.8)	0.448 *	0.366 *
T2 Total	28.1 (17.6)	0.399 *	0.427 *
Perceived threat of COVID-19			
T1 Perceived threat	13.1 (3.6)	−0.048	−0.091

* *p* < 0.001. Time 1: 12–15 gestational weeks; Time 2: 22–35 gestational weeks.

**Table 2 healthcare-11-00361-t002:** Correlations between the variables used in this study.

	1. Depression	2. Borderline Personality Traits	3. Adult Attachment	4. Perceived Impact of Pregnancy	5. W-DEQ	6. OCI-R	7. Perceived Threat
1.	---	0.412 *	0.271 *	−0.364 *	0.425 *	0.427 *	−0.174 *
2.	0.440 *	---	---	---	---	---	---
3.	0.306 *	0.431 *	---	---	---	---	---
4.	−0.452 *	−0.314 *	−0.278 *	---	---	---	---
5.	0.428 *	0.453 *	0.302 *	−0.465 *	---	0.427 *	0.023
6.	0.448 *	0.621 *	0.318 *	−0.240 *	0.465 *	---	0.009
7.	−0.094	−0.007	0.101	0.042	−0.003	0.061	---

* *p* < 0.001. Correlations at Time 1 are under the diagonal, whereas those at Time 2 are above the diagonal. Abbreviations: W-DEQ; Wijma Delivery Expectancy/Experience Questionnaire, OCI-R; Obsessive-Compulsive Inventory—Revised.

## Data Availability

The datasets used and analyzed in this study are available from the corresponding author upon reasonable request.

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
