# Peer review of "Relationships between Depression, Fear of Childbirth, and Obsessive-Compulsive Symptoms among Pregnant Women under the COVID-19 Pandemic in Japan"

_healthcare, 2023, doi:10.3390/healthcare11030361_

Round 1
Reviewer 1 Report
Thank you for the opportunity to review this manuscript.
Abstract
Include the setting and locations, and relevant dates, including periods of recruitment of data collection
The results could be better described in the abstract, presenting objective results and p values.
Main text
Methods
Could the authors inform if the scale Perceived threat of the COVID-19 is validated?
Results: Figure 1 and 2 are not clear, could the authors improve them? Some arrows does not connect elements. Please, I suggest clarify or exclude the figure. Correlation has statistical meaning, is it the correct word to be used in the caption?
The caption should better describe the figures. Include all the information needed to understand the figure.
Table 1 What means T1 depression and T2 depression? Do they represent the value of the AND scale or the answers to the items chosen and described in the methods? What means ‘primitive difences’, identity defusion’ and ‘reality testing’?
Could the authors include a table with descriptive data on the results of the scales used at different times? It would be interesting to understand the profile of the population.
Conclusions
I suggest to revise conclusions to present the main findings of this study and must be aligned with the objectives of the study.
Author Response
Manuscript ID: healthcare-2107554
"Correlates of antenatal depression with personality, adult attachment, fear of childbirth, obsessive-compulsive symptoms and perceived threat under the COVID-19 pandemic."
We are grateful to the editors and reviewers for careful reading our manuscript and for valuable comments. We revised the manuscript according to the reviewers’ comments. Please refer to detailed point-by-point responses as follows.
Point 1: Abstract include the setting and locations, and relevant dates, including periods of recruitment of data collection. The results could be better described in the abstract, presenting objective results and p values.
Response 1: We added the details of settings and results in the abstract as follows:
“Participants, who resides in Japan, were recruited were recruited for an Internet survey conducted in December 2020.”
“Structural equation modelling showed that AND predicted OCS (β = 0.16, p < 0.001), which in turn predicted FOC (β = 0.09, p = 0.042), and FOC in turn predicted AND (β = 0.23, p < 0.001).”
Point 2: Methods Could the authors inform if the scale Perceived threat of the COVID-19 is validated?
Response 2: Thank you for pointing it out. We added P3, lines141 blow;
“Perceived threat of the COVID-19: We used ad hoc questions.”
Point 3: Results Figure 1 and 2 are not clear, could the authors improve them? Some arrows does not connect elements. Please, I suggest clarify or exclude the figure. Correlation has statistical meaning, is it the correct word to be used in the caption?
The caption should better describe the figures. Include all the information needed to understand the figure.
Response 3: We reckon that the point you mentioned is our hiding error terms. Error terms were, of course, calculated but were hidden for visibility of the figure. This is why arrows look not to be connected. The two figures are, we think, are essential because they clearly show the model to start and the final trimmed model.
Additionally, the footnote in Table 2 was incorrectly shown as that in Figure 1, which we have corrected.
Point 4: Results Table 1 What means T1 depression and T2 depression? Do they represent the value of the AND scale or the answers to the items chosen and described in the methods? What means ‘primitive difences’, identity defusion’ and ‘reality testing’?
Response 4: Table 1 shows correlations of variables with T1 and T2 depression which measures by two items described in the methods. Primitive defence (PD), identity diffusion (ID), and reality testing (RT) disturbance are subscales of IPO-SV which measures borderline personality traits (As it was mentioned in Methods P3, lines 116 to 119).
Point 5: Results Could the authors include a table with descriptive data on the results of the scales used at different times? It would be interesting to understand the profile of the population.
Response 5: Please see Table 1. Means and SDs of each scale measured at T1 and T2 are shown.
Point 6: Conclusions I suggest to revise conclusions to present the main findings of this study and must be aligned with the objectives of the study.
Response 6: Thank you for your suggestion. We mentioned our findings for the primary objective, but not for the other question. We modified the conclusions as follows:
“By the means of a non-recursive SEM among Japanese pregnant women we aimed to understand the causal relationships between borderline personality traits and adult attachment with FOC, OCS, and AND as well as circular influences from AND through OCS to FOC, in turn, predicting AND. Although, no association was found between AND and the perception of COVID-19 threat.”

Reviewer 2 Report
Thank you for the opportunity to review this manuscript: "Correlates of antenatal depression with personality, adult attachment, fear of childbirth, obsessive-compulsive symptoms and perceived threat under the COVID-19 pandemic".
In my opinion, the article is well-written, clear, and logical. I appreciate the scientific soundness and quality of the presentation.
The subject matter addressed in the article is important. It allows for the development of knowledge about the mental health of pregnant women during COVID-19.
Below, I present issues that, in my opinion, require improvement or clarification. I show them the following parts of the manuscript.
The title of the article needs to be changed. Researchers are not only studying the correlates but want to looking for the determinants of depression.
Providing data from statistical analyzes at the beginning of the introduction does not seem to be a good solution (lines 29-30).
In the context of the researched issues, it would be important in the introduction to describe the specifics of prenatal care and the changes related to childbirth that took place in Japan during the pandemic waves in which the study was conducted.
The introduction lacks theoretical justification with reference to the literature, which justifies considering borderline personality traits, clinical variables, and relational traits, such as attachment styles. It needs deepening.
Emphasizing a nuclear emergency (lines 71-78) seems too distant a comparison. There are a lot of publications about the importance of the COVID-19 pandemic for mental health, including pregnant women, e.g.
Lobel, M., Preis, H., Mahaffey, B., Schaal, N. K., Yirmiya, K., Atzil, S., Reuveni, I., Balestrieri, M., Penengo, C., Colli, C., Garzitto, M., Driul, L., Ilska, M., Brandt-Salmeri, A., Kołodziej-Zaleska, A., Caparros-Gonzalez, R. A., Castro, R. A., La Marca-Ghaemmaghami, P., & Meyerhoff, H. (2022). Common model of stress, anxiety, and depressive symptoms in pregnant women from seven high-income Western countries at the COVID-19 pandemic onset. Social science & medicine (1982), 315, 115499. https://doi.org/10.1016/j.socscimed.2022.115499
The authors should describe the maintenance of ethical standards toward the subjects. They declare that the study was anonymous; it seems that they should explain how the anonymity of the respondents was preserved, combining the results of the respondents in points t1 and t2. How they provided the electronic money since the study was anonymous?
In addition, how is anonymity maintained? The authors write: All participants provided electrical informed consent after understanding the study rationale and procedure (lines 162-163). These inaccuracies require clarification.
The authors described Data analysis in detail but did not write what statistical programs were used.
The authors should change subscribers in Figure 1. Conceptual model of causal relationships between antenatal depression. χ2, df, CFI, and RMSEA are unreadable in the figure. It was changed in some strange signs.
Figure 2. Final model after trimming should appear in the section Results. In my opinion, the first figure is unnecessary. It looks like Figure 2, only without data. Maybe authors could present the model of research in a more complex and readable way.
Sentences (lines 168-171): 'As compared to the women who declined to participate in the T2, those who participated in the T2 survey were slightly but significantly older (32.2 vs. 31.4 years, p < .05) and their gestational age was less advanced (13.2 vs. 13.4, p < .05). The two groups of women did not significantly differ otherwise' are difficult to understand. Maybe it is not the appropriate place to put this information. The better place is the description of the procedure, inclusion and exclusion criteria, the process of study selection, and the missing surveys (before the results).
The Results part needs to be reorganized. The statistics presented in table 1 and 2, and their description should appear before figure 1. They are much simpler, and it is important to know them before making complex analyses.
It is unclear how the total score of adult attachment (in table 1 and in table 2) was calculated (it compose of the different styles of attachment and the total score is impossible to obtain)—similarly, Perception of the impact of the current pregnancy.
The correlations in table 2 are complicated for the reader to follow; they should appear diagonally.
How will the authors explain the relationships presented in the model, bearing in mind that such a low percentage of the surveyed women actually experienced symptoms of depression?
However, I wouldn't assume that pregnant women seemed to be immune to the threat of infection (lines 224-225). Maybe it is a problem of measurement? See: Preis, H., Mahaffey, B., & Lobel, M. (2020). Psychometric properties of the Pandemic-Related Pregnancy Stress Scale (PREPS). Journal of psychosomatic obstetrics and gynaecology, 41(3), 191–197. https://doi.org/10.1080/0167482X.2020.1801625
Research shows that the phenomenon's scale for pregnant women was huge in many countries (even with a small number of cases). The sense of threat and fear was influenced by changes in maternity care, isolation, inability to access fresh air, or lack of knowledge about the fetus being infected with the SARS-CoV-2 virus.
You write about this in the limitation of the study (lines 247-248). I think it is worth combining these threads and not underestimating the pandemic's impact on pregnant women's mental health.
With great respect for your work.
Author Response
Manuscript ID: healthcare-2107554
"Correlates of antenatal depression with personality, adult attachment, fear of childbirth, obsessive-compulsive symptoms and perceived threat under the COVID-19 pandemic."
We are grateful to the editors and reviewers for careful reading our manuscript and for valuable comments. We revised the manuscript according to the reviewers’comments. Please refer to detailed point-by-point responses as follows.
Point 1: The title of the article needs to be changed. Researchers are not only studying the correlates but want to looking for the determinants of depression.
Response 1: Following the review comments, we renamed the title “Relationships between depression, fear of childbirth, and obsessive-compulsive symptoms among pregnant women under the COVID-19 pandemic in Japan”.
Point 2: Providing data from statistical analyzes at the beginning of the introduction does not seem to be a good solution (lines 29-30).
Response 2: We deleted the statistical data in brackets (p.1, lines 32).
“The pooled prevalence of AND identified by a meta-analysis was 20.7% (95% CI 19.4–21.9%, p < 0.001), and that of major depression was 15% (95% CI 13.6–16.3%, p < 0.001) [1]. ”
Point 3: In the context of the researched issues, it would be important in the introduction to describe the specifics of prenatal care and the changes related to childbirth that took place in Japan during the pandemic waves in which the study was conducted.
Response 3: Following the reviewer’s comment, we added the sentence as followed (p.2, lines 70-72);
“In Japan, most obstetric facilities canceled or restricted perinatal classes with the COVID-19 pandemic, thus pregnant women were less likely to get support from healthcare providers.”
Point 4: The introduction lacks theoretical justification with reference to the literature, which justifies considering borderline personality traits, clinical variables, and relational traits, such as attachment styles. It needs deepening.
Response 4: Thank you for your comments. We revised the sentence and added other references as following; (p.1-2, lines 37-47)
“It is of note that these variables are not mutually independent. Out of the correlates, poor relationship with the partner or being exposed to domestic violence would be strongly contributing on the onset of perinatal depression [2,3]. Considering pregnancy is a life stage in which interpersonal relationships, particularly with a partner playing a critical role preparing for childbirth and child rearing [6]. The person pregnant women seek support most is a partner [7] and the partner’s support contributes mental health of pregnant women [8]. Although poor interrelationship is caused by malfunctional interaction between individuals with different personality [9], it is speculated that a woman with insecure attachment style would underlie the poor relationship among the couple, leading to the occurrence of AND.”
Point 5: Emphasizing a nuclear emergency (lines 71-78) seems too distant a comparison. There are a lot of publications about the importance of the COVID-19 pandemic for mental health, including pregnant women, e.g.
Lobel, M., Preis, H., Mahaffey, B., Schaal, N. K., Yirmiya, K., Atzil, S., Reuveni, I., Balestrieri, M., Penengo, C., Colli, C., Garzitto, M., Driul, L., Ilska, M., Brandt-Salmeri, A., Kołodziej-Zaleska, A., Caparros-Gonzalez, R. A., Castro, R. A., La Marca-Ghaemmaghami, P., & Meyerhoff, H. (2022). Common model of stress, anxiety, and depressive symptoms in pregnant women from seven high-income Western countries at the COVID-19 pandemic onset. Social science & medicine (1982), 315, 115499. https://doi.org/10.1016/j.socscimed.2022.115499
Response 5: As the reviewer commented, there are many papers focusing on stress resulting from the COVID-19 pandemic. However, we measured threat of the COVID-19 and examined the association with AND, because it is known that the perceived threat has a significant impact on mental health problems during disasters.
Point 6: The authors should describe the maintenance of ethical standards toward the subjects. They declare that the study was anonymous; it seems that they should explain how the anonymity of the respondents was preserved, combining the results of the respondents in points t1 and t2. How they provided the electronic money since the study was anonymous?
In addition, how is anonymity maintained? The authors write: All participants provided electrical informed consent after understanding the study rationale and procedure (lines 162-163). These inaccuracies require clarification.
Response 6: We agree with you. We added the following text in methods (p.2-3, lines 96-98).
“Participants were asked to enter their email address after completing the survey to merge the data of T1 and T2 only by the means of mail address. The participants were incentivized with electronic money points usable for Amazon shopping via the MTI Ltd.”
Point 7: The authors described Data analysis in detail but did not write what statistical programs were used.
Response 7: We added the following text in Data analysis (p.4, lines 164-166).
“Statistical analyses were performed using IBM SPSS version 28.0 and Amos version 28.0 software for Windows (IBM Corp., Armonk, NY).”
Point 8: The authors should change subscribers in Figure 1. Conceptual model of causal relationships between antenatal depression. χ2, df, CFI, and RMSEA are unreadable in the figure. It was changed in some strange signs. Figure 2. Final model after trimming should appear in the section Results. In my opinion, the first figure is unnecessary. It looks like Figure 2, only without data. Maybe authors could present the model of research in a more complex and readable way.
Response 8: Figure 1 shows the conceptual model to start with. Hence this figure is essential without which reader could not comprehend the results. As you pointed out, we inserted Figure 2 in the Results section.
Point 9: Sentences (lines 168-171): 'As compared to the women who declined to participate in the T2, those who participated in the T2 survey were slightly but significantly older (32.2 vs. 31.4 years, p < .05) and their gestational age was less advanced (13.2 vs. 13.4, p < .05). The two groups of women did not significantly differ otherwise' are difficult to understand. Maybe it is not the appropriate place to put this information. The better place is the description of the procedure, inclusion and exclusion criteria, the process of study selection, and the missing surveys (before the results).
Response 9: We moved the part the reviewer pointed out to the Methods section (p.3, lines 100-104).
Point 10: The Results part needs to be reorganized. The statistics presented in table 1 and 2, and their description should appear before figure 1. They are much simpler, and it is important to know them before making complex analyses.
It is unclear how the total score of adult attachment (in table 1 and in table 2) was calculated (it compose of the different styles of attachment and the total score is impossible to obtain)—similarly, Perception of the impact of the current pregnancy.
The correlations in table 2 are complicated for the reader to follow; they should appear diagonally.
Response 10: As we noted in Response 8, Figure 1 shows the conceptual model, thus it should be in Methods section.
We clearly mentioned how to calculate the total score of adult attachment in the methods (please see p.3, lines 120 to 125). This is proposed by the original authors.
The footnote in Table 2 was incorrectly shown as that in Figure 1, which we have corrected.
Point 11: How will the authors explain the relationships presented in the model, bearing in mind that such a low percentage of the surveyed women actually experienced symptoms of depression?
Response 11: As you pointed out, the prevalence of MDE was not high, though, many pregnant women had AND, FOC and OCS. This study identified the relationship between those three variables and the factors affecting them, and it would be useful for considering intervention for pregnant women.
Point 12: However, I wouldn't assume that pregnant women seemed to be immune to the threat of infection (lines 224-225). Maybe it is a problem of measurement? See: Preis, H., Mahaffey, B., & Lobel, M. (2020). Psychometric properties of the Pandemic-Related Pregnancy Stress Scale (PREPS). Journal of psychosomatic obstetrics and gynaecology, 41(3), 191–197. https://doi.org/10.1080/0167482X.2020.1801625
Research shows that the phenomenon's scale for pregnant women was huge in many countries (even with a small number of cases). The sense of threat and fear was influenced by changes in maternity care, isolation, inability to access fresh air, or lack of knowledge about the fetus being infected with the SARS-CoV-2 virus.
You write about this in the limitation of the study (lines 247-248). I think it is worth combining these threads and not underestimating the pandemic's impact on pregnant women's mental health.
Response 12: We totally agree with you, so we have already discussed the limitations of the measurement for the perceived threat of the COVID-19 and considered regarding assessment points.

Reviewer 3 Report
The paper presented for review includes the results of a study on antenatal depression. This is a disorder with an incompletely understood aetiology. In their study, the authors searched for a relationship between antenatal depression and childbirth anxiety and obsessive-compulsive symptoms, as well as the perceived threat of COVID-19. For this purpose, they `constructed a model of the relationship between the examined factors and depressive symptoms. The results obtained indicate a good fit of the model and thus the possibility of statistical inference about the co-occurrence of the variables studied.
The study was designed properly and the psychological measurement tools used were selected appropriately.
The only doubts are raised by the authors' overuse of the statement that the examined factors IMPACT. The study is done in the correlational trend and from a methodological point of view, we can only talk about co-occurrence of variables and we are not entitled to state an impact. We can only talk about impact when the study is done in an experimental design.
"Insecure adult attachment 21 influenced AND and FOC via the perceived negative impact of the current pregnancy."
Apart from this methodological remark, I think that the paper is interesting and qualifies for publication after the correction concerning Impact/Co-occurrence
Author Response
Manuscript ID: healthcare-2107554
"Correlates of antenatal depression with personality, adult attachment, fear of childbirth, obsessive-compulsive symptoms and perceived threat under the COVID-19 pandemic."
We are grateful to the editors and reviewers for careful reading our manuscript and for valuable comments. We revised the manuscript according to the reviewers’comments. Please refer to detailed point-by-point responses as follows.
Point 1: The only doubts are raised by the authors' overuse of the statement that the examined factors IMPACT. The study is done in the correlational trend and from a methodological point of view, we can only talk about co-occurrence of variables and we are not entitled to state an impact. We can only talk about impact when the study is done in an experimental design.
"Insecure adult attachment 21 influenced AND and FOC via the perceived negative impact of the current pregnancy."
Apart from this methodological remark, I think that the paper is interesting and qualifies for publication after the correction concerning Impact/Co-occurrence
Response 1: As the reviewer's comment, adult attachment, AND, and FOC were measured at the same point. However, it is common to use the expression that anchor items, such as adult attachment and personality, "predict" or "influence" (statistical jargons) other variables in SEM.

Round 2
Reviewer 1 Report
I am happy to recommend acceptance. I think the authors have adequately addressed the reviewers comments/suggestions.
Author Response
Manuscript ID: healthcare-2107554
"Correlates of antenatal depression with personality, adult attachment, fear of childbirth, obsessive-compulsive symptoms and perceived threat under the COVID-19 pandemic." (We renamed the title “Relationships between depression, fear of childbirth, and obsessive-compulsive symptoms among pregnant women under the COVID-19 pandemic in Japan” following the review comments).
We appreciate you for giving us the opportunity to strengthen our manuscript with your valuable and insightful comments.
We revised the manuscript according to the editors’ comments. Please find the comments and answers below.
Editor
Point 1: Correlation studies are meant to see relationships- not influence- even if there is a positive correlation between x and y, one can never conclude if x or y is the reason for such correlation. It can never determine which variables have the most influence. Thus, as recommended by reviewer 3, the caution is needed to re-word for the use of word "influence".
Response 1: We added the following text in Discussion (p.4, lines 207–212).
Use of a non-recursive model in which OCS, FOC, and depression were set to statistically predict each other was unique to this study. It is of note that our study examined correlations between the target variables at one time point as well as between the two time points, we used ‘influence’ and ‘predict’ only as statistical terms but did not necessarily indicate real causality. We cannot more cautious about unseen confounders and mediators.

Reviewer 2 Report
Thank you for the opportunity to review the article again. In my opinion, the introduced changes and clarifications are sufficient. To increase the scientific quality of the text, I would also suggest entering the reliability of each tool used in the study (e.g., Cronbach's alpha). You may consider it.
Author Response
Manuscript ID: healthcare-2107554
"Correlates of antenatal depression with personality, adult attachment, fear of childbirth, obsessive-compulsive symptoms and perceived threat under the COVID-19 pandemic." (We renamed the title “Relationships between depression, fear of childbirth, and obsessive-compulsive symptoms among pregnant women under the COVID-19 pandemic in Japan” following the review comments).
We appreciate you for giving us the opportunity to strengthen our manuscript with your valuable and insightful comments. We revised the manuscript according to the editors’ reviewers’ comments. Please find all the comments and answers below.
Point 1: To increase the scientific quality of the text, I would also suggest entering the reliability of each tool used in the study (e.g., Cronbach's alpha).
Response 1: Following the reviewer’s comment, we calculated Cronbach's alpha and added the sentence in Methods as followed;
(p.3, lines 115–116) “In this study, Cronbach's α coefficients were 0.83 at T1, and 0.79 at T2.”
(p.3, lines 120–121) “In this study, Cronbach’s α coefficients of the PD, ID, and RT subscales were 0.71, 0.78, and 0.87, respectively.”
(p.3, lines 127–128) “In this study, Cronbach’s α coefficients were 0.65.”
(p.3, lines 139) “In this study, Cronbach's α coefficients were 0.91 at T1, and 0.91 at T2.”
(p.3, lines 143–144) “In this study, Cronbach's α coefficients were 0.89 at T1, and 0.90 at T2.”
(p.3, lines 148) “In this study, Cronbach’s α coefficients were 0.67.”
Editor
Point 1: Correlation studies are meant to see relationships- not influence- even if there is a positive correlation between x and y, one can never conclude if x or y is the reason for such correlation. It can never determine which variables have the most influence. Thus, as recommended by reviewer 3, the caution is needed to re-word for the use of word "influence".
Response 1: We added the following text in Discussion (p.4, lines 207–212).
Use of a non-recursive model in which OCS, FOC, and depression were set to statistically predict each other was unique to this study. It is of note that our study examined correlations between the target variables at one time point as well as between the two time points, we used ‘influence’ and ‘predict’ only as statistical terms but did not necessarily indicate real causality. We cannot more cautious about unseen confounders and mediators.
